# Effect of Natural Disaster-Related Prenatal Maternal Stress on Child Development and Health: A Meta-Analytic Review

**DOI:** 10.3390/ijerph18168332

**Published:** 2021-08-06

**Authors:** Sandra Lafortune, David P. Laplante, Guillaume Elgbeili, Xinyuan Li, Stéphanie Lebel, Christian Dagenais, Suzanne King

**Affiliations:** 1Department of Psychology, Faculty of Arts and Science, University of Montreal, Marie-Victorin Pavilion, Montreal, QC H2V 2S9, Canada; sandra.lafortune@umontreal.ca (S.L.); stephanie.lebel@umontreal.ca (S.L.); christian.dagenais@umontreal.ca (C.D.); 2Lady Davis Institute for Medical Research, Jewish General Hospital, Montreal, QC H3T 1E2, Canada; david.laplante@ladydavis.ca; 3Mental Health and Society Division, Douglas Hospital Research Centre, Perry Pavilion, Montreal, QC H4H 1R3, Canada; guillaume.elgbeili@douglas.mcgill.ca; 4Department of Psychiatry, Faculty of Arts and Science, McGill University, Ludmer Research & Training Building, Montreal, QC H3A 1A1, Canada; xinyuan.li2@mail.mcgill.ca

**Keywords:** prenatal maternal stress, natural disasters, child development, meta-analysis

## Abstract

The evidence supporting the idea that natural disaster-related prenatal maternal stress (PNMS) influences the child’s development has been accumulating for several years. We conducted a meta-analytical review to quantify this effect on different spheres of child development: birth outcomes, cognitive, motor, physical, socio-emotional, and behavioral development. We systematically searched the literature for articles on this topic (2756 articles retrieved and 37 articles included in the systematic review), extracted the relevant data to calculate the effect sizes, and then performed a meta-analysis for each category of outcomes (30 articles included across the meta-analyses) and meta-regressions to determine the effect of some factors of interest on the association between PNMS and child development: type of PNMS (objective, psychological, cognitive, diet), type of natural disaster (ice storm, flood/cyclone), type of report (maternal, third-party observer, medical), timing of exposure (preconception exposure included or not) and child age at assessment (under 10 or 10 years and older). We found that PNMS significantly influences all spheres of child development. Higher PNMS levels were associated with longer gestational age, larger newborns, and higher BMI and adiposity levels, as well as worse cognitive, motor, socio-emotional, and behavioral outcomes.

## 1. Introduction

Evidence suggests that prenatal maternal stress (PNMS) is associated with suboptimal development in children [1]. In the PNMS literature, “stress” is a broadly defined concept encompassing anxiety, depression, and exposure to stressful life events. The severity of the consequences for prenatally exposed children differs depending on the type of stressor being studied. For example, the association between PNMS and socioemotional problems is stronger for prenatal maternal depression (OR = 1.79; 95% CI = 1.61–1.99) than for prenatal maternal anxiety (OR = 1.50; 95% CI = 1.36–1.64) [2]. Another form of maternal stress that has been studied in recent years, and is associated with global warming, is natural disasters. However, the overall magnitude of the effects of disaster-related PNMS on child outcomes has yet to be calculated.

In order to specify our research topic, it is necessary to make the distinction between natural disasters, climate change, and man-made disasters (e.g., terrorism, wars, nuclear hazards, and oil spills), all of which have been shown to be linked to suboptimal child development [3]. Climate change has impacted (primarily negatively) the lives of millions of people worldwide [4], is believed to be associated with increased heat-related mortality, morbidity, and infectious diseases [5], and has undoubtably contributed to the recent rapid rise in the frequency and intensity of natural disasters (e.g., heatwaves, floods, tropical cyclones) occurring over recent decades [6]. That being said, climate change is more of an aggravating factor rather than a *sine qua non* condition for the occurrence of natural disasters.

Globally, natural disasters affect humankind in terms of mortality, injury, and displacement [7,8,9,10,11]. The World Health Organization refers to a natural disaster as “an act of nature of such magnitude as to create a catastrophic situation in which the day-to-day patterns of life are suddenly disrupted and people are plunged into helplessness and suffering and, as a result, need food, clothing, shelter, medical and nursing care and other necessities of life, and protection against unfavourable environmental factors and conditions” [12] (p. 14). A natural disaster can also be conceptualized as an “extreme event as any manifestation in a geophysical system (lithosphere, hydrosphere, biosphere or atmosphere) which differs substantially or significantly from the mean” that occurs when “human socio-economic and physiological systems do not have the capacity sufficiently to reflect, absorb or buffer the impact” [13]. In brief, a natural disaster is a sudden and acute event generated by one of the earth’s geophysical systems that disrupts a population to the point of exceeding its management capacities.

Men and women are differentially affected by natural disasters: younger, less educated, and poorer women are the most affected group [14]. A recent systematic review also found that violence against women and girls increases in the aftermath of a natural disaster, compounding the effect of the disaster itself [15]. The repercussions of disasters on women can go beyond their own experiences. During pregnancy, the prenatal environment of the fetus (i.e., the mother’s womb and the placenta) relays signals to the unborn child about the external world and can generate a “predictive adaptative response” [16]. The fetus thus develops in a manner that would help to assure its immediate survival in this predicted postnatal world, although perhaps to the detriment of long-term health and well-being. According to the Developmental Origins of Health and Disease (DOHaD) hypothesis [17], a mismatch between the prenatal and postnatal environments can put the unborn child at risk of developing problems; the prenatal environment can alter genome expression that is no longer adapted to the postnatal environment. As such, in utero exposure to disaster related PNMS could prepare the child to live in a stressful environment that is potentially no longer representative of the actual postnatal environment. Thus, prenatal exposure to a temporary stressor can result in a permanent, ill-fitting phenotype in the postnatal world.

Several researchers have been studying the consequences of PNMS in the context of natural disasters on the development of the child. Disasters can produce natural experiments with quasi-experimental designs, given that these kinds of events often distribute their hardship in a quasi-random manner in the population. A significant advantage of this research design is that it allows one to study components of stress that are independent of the parents’ personal attributes such as their genetics, personality, and propensity to create psychosocial stress.

Studying disaster-related PNMS also allows investigators to determine which aspect(s) of the pregnant woman’s disaster experiences are related to child outcomes. The pregnant woman’s stress experience can be divided into separate components [18]: the degree of objective hardship experienced, the severity of psychological distress, and the cognitive appraisal of the crisis, in addition to the woman’s physiological response [19]. These different aspects of PNMS could lead to a large spectrum of consequences in the child, including problems associated with cognitive, behavioral, motor, and/or physical development. The intensity of the pregnant woman’s response, the timing within pregnancy of exposure to the onset of the PNMS (i.e., preconception, 1st, 2nd, or 3rd trimester of pregnancy) [1], and sex of the child [20] may result in differential effects on the child. Knowing the moment in pregnancy during which the fetus is exposed to PNMS is important because fetal development evolves throughout pregnancy, creating “windows of vulnerability” in gestation [21,22].

### Study Aims

The goal of this project was to conduct a meta-analysis for each category of outcomes using the available data in order to estimate the magnitude of the effects of disaster-related PNMS (exposure before conception or in utero) on different spheres of child development from birth to late adolescence.

## 2. Materials and Methods

### 2.1. Protocol

The protocol for this meta-analytic review has been deposited in PROSPERO (https://www.crd.york.ac.uk/prospero/display_record.php?RecordID=152724, accessed on 27 April 2021).

### 2.2. Search Strategy

MEDLINE, EMBASE, Web of Science, CINAHL, and PsycINFO were searched from the date of their inception to June 2020. Search strategies (Appendix A) were constructed with a range of text words and indexed terms related to “prenatal”, “natural disasters”, and “neonate/child/adolescent”. MeSH terms were used for each one of these constructs. In MEDLINE, for “natural disasters”, we used the following terms: Natural Disasters; Cold Temperature; Hot Temperature; Volcanic Eruptions; Snow. For “prenatal”, the following terms were used: Embryonic Structures; Fetus; Pregnant Women; Pregnancy; Maternal Exposure; Prenatal Exposure Delayed Effects. For “neonate/child/adolescent”, the following terms were used: infant; child; adolescent. For other databases, the appropriate equivalent subject headings were included in the search strategies, as shown in Appendix A.

### 2.3. Study Eligibility Criteria

To be included in the meta-analytic review, studies had to meet several eligibility criteria. Only published, peer-reviewed, empirical studies were included. The target population had to be humans less than 18 years of age. The children had to be exposed before conception or in utero to natural disaster related PNMS. While climate change and man-made disasters can result in human tragedy and potentially influence fetal development, these events will not be covered in the present review. As such, studies referring only to temperature (e.g., season, ambient temperature, heat waves, cold spells, extreme weather) have been excluded from this review, since these are not well-defined events, but also because there is a lack of clear consensus about the degree of variation from the mean, or the duration that such an event should last, in order to be considered potentially harmful to the exposed population. We also excluded any study referring to pollution per se, since it could be the result of human activity. Made-made disasters (nuclear reactor meltdowns, war, terrorist attacks) have also been excluded. Finally, biologic disasters, such as epidemics, have also been excluded. The current COVID-19 pandemic reminds us of the importance of future research on that topic.

We only included studies published in English, French, or Chinese since members of our team were fluent in these languages. We used an individual-level psychological approach, rather than a population-level epidemiological approach: we were interested in the individual experience of stress during pregnancy and, thus, a direct measure (e.g., physiological measures, self-report) of PNMS in pregnant women was required. In terms of outcomes, since most studies in this area are carried out on community samples rather than at-risk populations, the outcomes are primarily traits and/or variations from the mean rather than clinical diagnoses. For example, since schizophrenia affects only a small proportion of the population, the study of risk factors for the psychosis continuum in the community can be more informative [23]. It is for this reason that we decided to include only outcomes that are reported on a continuum. Finally, this review uses a clinical, rather than a fundamental, research perspective and its aim is not to establish the biological mechanisms that explain how PNMS appears to influence the various outcomes.

### 2.4. Study Selection

Covidence (https://www.covidence.org/ accessed on 22 July 2020), an online screening software, was used to screen all identified studies for inclusion by three of the authors (SLaf, DPL, and SLeb). First, all studies were imported into Covidence, and duplicates were removed. Second, titles and abstracts were screened independently by two authors for inclusion. Conflicts were discussed with the third author until a consensus was reached. Inter-rater reliability for titles and abstract screening was substantial [24] (κ = 0.670 (95% CI, 0.668 to 0.671), *p* < 0.001). Third, the remaining studies were assessed for full-text eligibility independently by two authors. Again, conflicts were discussed with the third author until a consensus was obtained. Inter-rater reliability for full-text review was moderate (κ = 0.462 (95% CI, 0.458 to 0.465), *p* < 0.001). Finally, the reference list from all included studies were searched for supplemental studies, but none were found. When the same findings were reported in different articles, we contacted the authors to verify the presence of duplicate data reporting. Only data from articles with the largest sample sizes were extracted and used for our analyses.

### 2.5. Data Extraction

The following data were extracted from the included studies: author, date of publication, country of the disaster, study design, type of natural disaster, prenatal exposure period, age of child at assessment, sample size, PNMS measures, outcomes, and effect sizes (correlation coefficients or difference in means).

The predictor of interest in this review is disaster-related PNMS. The primary developmental outcomes of interest are as follows: birth outcomes and cognitive, motor, physiological, physical, socio-emotional, and/or behavioral development. We also considered the following effect measures: PNMS type (maternal objective hardship, psychological distress, cognitive appraisal, diet change and cortisol), age of the child at assessment (under 10 years, or 10 years and older), timing of exposure in pregnancy (preconception or in utero), type of natural disaster (ice storm, flood/cyclone, or earthquake), and whether the outcomes of interest were maternal-rated, third-party observer rated, or from a medical report.

Objective hardship was defined as events experienced by the women that were independent of the parents’ characteristics, such as their temperament or judgment, and that could be quantified [25]. Psychological distress was defined as the women’s psychological reaction to the disaster [26]. Cognitive appraisal was defined as the maternal evaluation of the valence of the consequences of the event on her and her household [27]. Diet change was defined as any alteration in food accessibly and/or consumption due to the disaster. Finally, any measures of disaster-related cortisol functioning (baseline, area under the curve, waking response) were also included.

The only outcomes extracted were the ones defined as such by the authors. Meta-data and results were collected from the records in a systematic way using a form developed for this specific meta-analytic review. Data were extracted independently by two authors in order to limit errors, but also to minimize the risk of potential biases introduced by the authors.

### 2.6. Quality Assessment

Most studies included in our meta-analytic review were prospective longitudinal studies using a single cohort of children without a comparison group. Most quality assessment tools are made for randomized controlled trials while instruments developed for other designs have not yet demonstrated robustness [28]. Therefore, the quality assessment of the included studies was limited to judging the presence/absence of selective reporting bias by the three authors. Funnel plots and the trim and fill procedure [29] were used to detect and adjust for publication bias. To address the diversity of scales used in the PNMS field, we described the psychometric properties of the ones used in the included studies.

### 2.7. Statistical Analysis

Since we predicted high heterogeneity across studies because they featured different natural disasters, different PNMS scales, and different outcomes reported, we used a random-effect model to quantitatively synthesize data [30]. We used the DerSimonian & Laird estimator for heterogeneity variance, which we report for all analyses. Heterogeneity was defined according to Higgins’ definition: low (I^2^ = 25%), moderate (I^2^ = 50%), and high (I^2^ = 75%) [31]. Since the objective of this meta-analytic review was to determine the magnitude of the association between PNMS and child development, we extracted effect sizes reported in the studies. Because most studies reported correlation coefficients (*r*), for studies that reported differences in means we converted them to correlation coefficients.

We reversed the direction of some correlations to ensure that an increase in the predictor meant greater exposure to the stressor and that an increase in the outcome meant worse child development. Most of the birth outcomes we extracted were related to size at birth (e.g., birth weight, birth length) and none were adjusted by gestational age. For these measurements, a higher score suggests a larger newborn. To remain logical in our data grouping, we excluded two measures for which an increase did not equate with a larger newborn: head circumference to birth length ratio and average dermatoglyphic finger ridge count asymmetry. For some physiological outcomes, the meaning of their directionality has not yet been established (e.g., child cortisol). For others, it is a balance between indices rather than an absolute level that is recommended (e.g., c-peptide, cytokine production, testosterone levels). Consequently, we excluded these outcomes from analyses. We also excluded variables that were reported in terms of change rather than a measure at some point in time because it did not make sense to group them with the other measures. This is because an increase or decrease in change is relative to the start and endpoint, while a measurement taken at a specific point in time gives an indication of the condition of the individual. When the raw score and the standardized score were both reported for the same outcome, we used the raw score since the purpose of our analyses is not to compare data across samples, and because the characteristics of the participants vary from one sample to the other.

We then performed a Fisher Z transformation meta-analysis for each category of outcomes: all correlation coefficients were transformed to normally distributed variables to determine the confidence intervals before computing the average z-values and their confidence intervals and *p*-values, then retransforming the z-values back into r’s [32].

The six meta-analyses were conducted separately for each category of outcomes (birth outcomes, cognitive, motor, physical, socio-emotional, and behavioral) in order to derive clinically meaningful conclusions. We illustrate the results for each category in a forest plot showing the effect sizes that were extracted. We then performed sensitivity analyses to make sure that none of the results were pulled by an artifact, and that the results remained stable after the removal of each effect size individually. In order to detect a publication bias, the trim and fill procedure [29] was used to estimate the number of studies missing from each meta-analysis. We then represented the effect sizes in a funnel plot.

Next, for each meta-analysis, we performed meta-regression analyses to determine the extent to which different factors influenced the magnitude of the global association: natural-disaster PNMS effect (e.g., objective hardship, psychological distress, cognitive appraisal, diet change), age effect (under 10 years of age, vs. 10 years or older) [33], timing of exposure effect (preconception included or not), types of natural disaster effect (ice storm, flood/cyclone, or earthquake) and type of report effect (maternal report, third party observer (e.g., research team, teacher), or medical report). For the type of PNMS, if the individual scale scores were not available, we used the composite score. The criterion for performing a meta-regression was that each category of effects had to present at least five effect sizes. This criterion prevented us from comparing the effect of earthquakes to the other types of disaster since only four such effects sizes were extracted. Finally, we performed Fisher’s exact test of independence on all factors for each type of outcome. If two factors were not independent, we could not distinguish the effect of these two factors on the relationship between PNMS and child development.

## 3. Results

### 3.1. Database Search Process

Figure 1 presents the study selection flow chart. We identified 2756 records through database searching. After removing duplicates, we screened the remaining 1943 records by title and abstract, discarding irrelevant studies. We then assessed 342 full-text manuscripts using the list of eligibility criteria described earlier, resulting in 41 studies being included. However, four studies were excluded because they reported the same data as another included study, but with a smaller sample size.

### 3.2. Data Extraction

The 37 studies included in this review were all published in English between 2004 and 2020 [19,25,34,35,36,37,38,39,40,41,42,43,44,45,46,47,48,49,50,51,52,53,54,55,56,57,58,59,60,61,62,63,64,65,66,67,68]. Most of them were conducted in Australia (*n* = 14 reports), Canada (*n* = 13), or the United States of America (*n* = 6), while the remaining four studies were conducted in China (*n* = 1), Haiti (*n* = 1), Vanuatu (*n* = 1), and Thailand (*n* = 1). Seven natural disasters were studied: the 2011 Queensland Flood in Australia (*n* = 14), the 1998 Quebec Ice Storm in Canada (*n* = 13 reports), the 2008 Iowa Flood (*n* = 4) and 2009 Red River Flood (*n* = 2) in the U.S.A., the 2008 Sichuan Earthquake in China (*n* = 1), the 2010 Earthquake in Haiti (*n* = 1), the 2011 Flood in Thailand (*n* = 1), and the 2015 Cyclone Pam in Vanuatu (*n* = 1). Among the 37 studies, 31 were conducted by the Stress in Pregnancy International Research Alliance (SPIRAL; www.mcgill.ca/spiral accessed on 23 June 2021). This research program includes Project Ice Storm (1998 Quebec Ice Storm; *n* = 13 reports), the Iowa Flood Study (2008 Iowa Flood; *n* = 4), and the Queensland Flood Study (QF2011; *n* = 14). One study used a retrospective design [40] while the remaining thirty-six studies used prospective designs. The sample sizes ranged between 30 [37] and 857 children [41]. Twenty-eight studies did not include preconception cases, while the other nine studies included children born up to four years after the disaster [35]. The characteristics of studies included in the review are presented in Table 1. However, we excluded six studies [25,37,53,57,66,67] from the meta-analysis process because they only reported physiological outcomes and we removed another study reporting only average dermatoglyphic finger ridge count asymmetry [19]. The meta-analyses finally included a total of 30 articles.

### 3.3. Child Development Outcomes

#### 3.3.1. Birth Outcomes

This category included the following outcomes: birth weight, birth length, head circumference, ponderal index, and gestational age. For birth outcomes, a correlation greater than zero suggests that greater PNMS levels were associated with a larger newborn. Twenty-five effect sizes extracted from five different studies were included in this analysis (Figure 2). Overall, there was a significant positive association between PNMS and birth outcomes (*r* = 0.0547 (95% CI = [0.0256; 0.0836]; Z = 3.69; *p* = 0.0002)), such that greater PNMS levels were associated with larger newborns. Due to its large sample size (*n* = 857), one effect size associated with a single study [40] accounted for 18.8% of the total weight. However, the sensitivity analysis revealed that the overall effect remained significant when rerunning the analysis after removing this effect or removing any other single effect. We found low heterogeneity between the studies (I^2^ = 0.0% [0.0%; 36.0%]).We performed a trim and fill procedure [29] and found that three positive correlations would be required to adjust for publication bias; thus, three positive effect sizes were added and are represented by red triangles in Figure 3, which were the inverse of 3 negative effect sizes already in the analysis. After the addition of these three effect sizes, the association between PNMS and birth outcomes was even more significant (*r* = 0.0640 (95% CI = [0.0343; 0.0937]; Z = 4.21; *p* < 0.0001)). We still found low heterogeneity between the studies after this procedure (I^2^ = 7.6% [0.0%; 39.5%]).

We ran meta-regressions to test how different factors might explain the differences in the associations between PNMS and birth outcomes. First, we tested the effect of the different types of PNMS on birth outcomes: objective hardship (12 effect sizes); psychological distress (1 effect size); cognitive appraisal (0 effect size); and diet change (12 effect sizes). Only objective hardship and diet change were included in the analysis since the other factors did not meet the criterion of at least five cases per category. There was a significant positive overall correlation between the birth outcomes and both objective hardship (*r* = 0.0618; SE = 0.0207; *p* = 0.0028) and diet change (*r* = 0.0555; SE = 0.0216; *p* = 0.0102) such that both higher objective hardship and diet change were associated with larger newborns (Figure 4). The summary effects of objective hardship and diet change on the birth outcomes did not differ significantly (*p* = 0.8335).

Next, we ran a meta-regression to test the effect of including, or not including, the preconception cases. Nineteen effect sizes did not include preconception cases in their sample, while six did. We found a significant positive overall correlation between size at birth and PNMS in both no-preconception-included effect sizes (*r* = 0.0480; SE = 0.0173; *p* = 0.0055) and preconception-included effect sizes (*r* = 0.0737; SE = 0.029; *p* = 0.0110) (Figure 4). Including or not including preconception cases in the analyses did not make a difference in the association between PNMS and birth outcomes (*p* = 0.4457).

We could not run meta-regressions for birth outcomes to test the age effect (all outcomes taken at birth), the natural disaster effect (only floods/cyclones apart from one earthquake) nor the report effect (only medical reports).

The Fisher’s test revealed no significant dependence among the factors tested in the meta-regressions indicating that the results were not confounded, and that we can assume that the effect found with one factor is not attributable to another factor.

#### 3.3.2. Cognitive Outcomes

This category included the following outcomes: developmental quotient, Mental Development Index (mental scale of the Bayley Scales of Infant Development), IQ (Wechsler Preschool and Primary Scale of Intelligence), production and receptive language abilities (MacArthur-Bates Communicative Development Inventories), and play style. For this category of outcomes, a correlation above zero means that greater PNMS levels were associated with worse cognitive outcomes. We retrieved 58 effect sizes extracted from nine studies (Figure 5). Their combination resulted in a significant positive association between PNMS levels and the cognitive development of the child (*r* = 0.1206 (95% CI = [0.0710; 0.1696]; Z = 4.75; *p* < 0.0001)): greater PNMS levels were associated with worse cognitive development in children. We performed a sensitivity analysis and we found that the overall effect remained significant no matter which effect size was removed. We found moderate heterogeneity between the studies (I^2^ = 72.6% [64.5%; 78.9%]). We performed a trim and fill procedure [29] and found that no supplemental effect size would be required to adjust for publication bias (Figure 6).

We tested the effect of the different types of PNMS on the cognitive outcomes: objective hardship (19 effect sizes); psychological distress (33 effect sizes); cognitive appraisal (6 effect sizes); and diet change (0 effect size). We found a significant positive overall correlation between the cognitive outcomes and both objective hardship (*r* = 0.1682; SE = 0.0456; *p* = 0.0002) and psychological distress (*r* = 0.1178; SE = 0.0337; *p* = 0.0005) such that higher objective or psychological PNMS levels were associated with worse cognitive development (Figure 7). The association between cognitive appraisal and cognitive outcomes was not significant (*r* = 0.0082; SE = 0.0760; *p* = 0.9145). There was no significant difference between the summary effects of objective hardship and psychological distress (*p* = 0.3748), nor between objective hardship and cognitive appraisal (*p* = 0.0712), nor between psychological distress and cognitive appraisal (*p* = 0.1875).

We ran another meta-regression to test the effect of the type of natural disaster on the relationship between PNMS and cognitive development. The effect sizes were related to different natural disasters: 36 were related to a flood, 20 to an ice storm, and only two to an earthquake, which we did not include in the analysis. We found a significant positive overall correlation between cognitive outcomes and PNMS in the ice-storm-related effect sizes (*r* = 0.2389; SE = 0.0322; *p* < 0.0001); however, the overall correlation between PNMS and cognitive outcomes was not significant in the flood-related effect sizes (*r* = 0.0329; SE = 0.0176; *p* = 0.0617) (Figure 7). There was a significantly higher summary effect in the group of ice storm studies than in the flood group (*p* < 0.0001).

Next, we tested the effect of the type of report on the association between PNMS and cognitive development. This analysis included 23 effect sizes for which the outcome had been reported by the mother, 35 that had been reported by a third-party observer, and 0 that had been obtained from medical reports. The association between PNMS levels and cognitive development was significantly positive for third-party observer-reported effect sizes (*r* = 0.1607; SE = 0.0328; *p* < 0.0001) but was not significant for the mother-reported effect sizes (*r* = 0.0633; SE = 0.0395; *p* = 0.1091) (Figure 7). The comparison between the effect of the two types of report suggested no significant difference (*p* = 0.0579).

According to the Fisher’s test, there was a significant association between the type of report and the type of disaster (*p* = 0.0237). Most effect sizes from the ice storm study were reported by a third-party observer. It was then impossible to distinguish the effect of both these factors.

Finally, we could not run meta-regressions to test the timing of exposure effect (only two effect sizes including preconception cases) and the age effect (all effect sizes under age 10).

#### 3.3.3. Motor Outcomes

This category included the following outcomes: fine and gross motor functioning, balance (postural control), bilateral coordination, and visual-motor integration. For the motor outcomes, a positive correlation suggested that greater PNMS levels were associated with worse motor outcomes. Sixty-eight effect sizes extracted from five studies (Figure 8) were combined and resulted in a positive significant association between PNMS levels and motor outcomes (*r* = 0.0829 (95% CI = [0.0534; 0.1122]; Z = 5.50; *p* < 0.0001)), meaning that greater PNMS levels were associated with worse motor development. The sensitivity analysis suggested that none of the effect sizes included in the analysis was solely responsible for the correlation. We found low heterogeneity between the studies (I^2^ = 45.8% [27.8%; 59.3%]). We performed a trim and fill procedure [29] and found that 15 negative correlations would be required to adjust for publication bias. We added 15 negative effect sizes (red triangles in Figure 9), which were the inverse of 15 positive effect sizes already in the analysis. Even after the addition of these 15 effect sizes, the association between PNMS and motor outcomes was still significant (*r* = 0.0371 (95% CI = [0.0041; 0.0700]; Z = 2.20; *p* = 0.0275)). Heterogeneity between the studies was then moderate (I^2^ = 63.7% [54.2%; 71.3%]).

We performed a meta-regression to determine the effect of the types of PNMS on the motor outcomes: objective hardship (17 effect sizes); psychological distress (37 effect sizes); cognitive appraisal (14 effect sizes); and diet change (0 effect size). There was a significant positive overall correlation between the motor outcomes and both objective hardship (*r* = 0.0994; SE = 0.0308; *p* = 0.0013) and the psychological measures (*r* = 0.0827; SE = 0.0207; *p* < 0.0001): higher objective hardship or psychological distress levels were associated with worse motor outcomes (Figure 10). However, the overall correlation between cognitive appraisal and motor development was not significant (*r* = 0.0650; SE = 0.0334; *p* = 0.0518). The summary effect of the three types of PNMS measures on motor outcomes did not differ significantly (objective hardship vs. psychological distress (*p* = 0.6512); objective hardship vs. cognitive appraisal (*p* = 0.4492); psychological distress vs. cognitive appraisal (*p* = 0.6535)).

We then ran a meta-regression to test the effect of the type of natural disaster on the association between PNMS levels and motor outcomes. We retrieved 62 effect sizes related to a flood, six related to an ice storm, and none related to an earthquake. We found a significant positive overall correlation between motor outcomes and PNMS in both flood related effect sizes (*r* = 0.0741; SE = 0.0153; *p* < 0.0001) and ice storm related effect sizes (*r* = 0.1978; SE = 0.0548; *p* = 0.0003) (Figure 10). The summary effect found in the ice storm group was significantly higher than in the flood group (*p* = 0.0297).

Lastly, we tested the effect of the type of report for the outcome on the association between PNMS levels and motor development. The analysis included 42 maternally reported effect sizes, 26 effect sizes reported by a third-party observer; no effect sizes were associated with a medical report. There was a significant positive overall correlation between PNMS levels and motor development when the outcome was either reported by the mother (*r* = 0.0809; SE = 0.0195; *p* < 0.0001) or a third-party observer (r = 0.0865; SE = 0.0242; *p* = 0.0004) (Figure 10). We did not find a significant difference between the summary effect of these two types of reports (*p* = 0.8581).

Due to a failure to meet the pre-established criterion to perform our meta-regressions, we could not test the age effect (all effect sizes were for children under age 10) or the timing of exposure effect (no preconception-included effect sizes).

The Fisher’s test revealed a significant association between the type of report and the type of disaster (*p* = 0.0021). For this category of outcomes, effect sizes extracted from ice storm studies were only reported by a third-party observer, while effect sizes extracted from flood studies were mostly reported by the mother. This made it impossible for us to conclude whether the effect observed with these two factors was due to one factor or the other.

#### 3.3.4. Physical Outcomes

Since this category included outcomes such as body mass index and adiposity, a correlation higher than zero suggested that greater PNMS levels were associated with children who are heavier relative to their height. Forty-two effect sizes extracted from three studies (Figure 11) were combined and resulted in a significantly positive association between PNMS levels and physical outcomes (*r* = 0.1040 (95% CI = [0.0585; 0.1490]; Z = 4.47; *p* < 0.0001)), such that greater PNMS levels were associated with children who are heavier relative to their height. The sensitivity analysis suggested that the overall effect remained significant no matter which effect size was removed. We found low heterogeneity between the studies (I^2^ = 39.7% [12.4%; 58.4%]). The trim and fill procedure [29] revealed that two negative correlations would be required to adjust for publication bias. After the addition of two negative effect sizes (red triangles in Figure 12), the association between PNMS and physical outcomes remained significant (*r* = 0.0925 (95% CI = [0.0454; 0.1391]; Z = 3.84; *p* = 0.0001)). Heterogeneity between the studies was low after this procedure (I^2^ = 45.2% [21.7%; 61.6%]).

We performed meta-regressions to test the effect of the different types of PNMS on the physical outcomes. PNMS measures were distributed as follows: objective hardship (16 effect sizes); psychological distress (16 effect sizes); cognitive appraisal (10 effect sizes); and diet change (0 effect sizes). In the case of maternal cortisol, because the significance of its directionality is not yet established, and because different cortisol measures may have different meanings, which prevented their grouping for analyses, we did not include this indicator of stress in our analyses. There was a significant positive overall correlation between the physical outcomes and both objective hardship (*r* = 0.1539; SE = 0.0299; *p* < 0.0001) and psychological distress (*r* = 0.1532; SE = 0.0299; *p* < 0.0001) such that higher objective hardship or psychological distress levels were associated with children who are heavier relative to their height (Figure 13). However, a non-significant overall correlation between negative cognitive appraisal levels and physical outcomes was observed (*r* = −0.0674; SE = 0.0383; *p* = 0.0780), with the negative direction of the effect indicating a tendency for a negative maternal cognitive appraisal to be associated with lower adiposity. Given that the direction of the association between cognitive appraisal and physical outcomes differed from those observed for objective hardship and subjective distress, there was a significant difference between the summary effect of cognitive appraisal on physical outcomes and that of both objective (*p* < 0.0001) and psychological measures (*p* < 0.0001). There was no significant difference between the summary effect of the objective and psychological measures (*p* = 0.9866).

We then performed a meta-regression to determine the effect of the type of natural disaster on the relationship between PNMS and physical development. We found 15 effect sizes related to a flood and 27 related to an ice storm. The overall correlation between PNMS levels and physical development was significantly positive when the disaster experienced by the mother was an ice storm (*r* = 0.1316; SE = 0.0293; *p* < 0.0001), but not when it was a flood (*r* = 0.0597; SE = 0.0374; *p* = 0.1105) (Figure 13). The difference between the summary effect of the two types of natural disasters was, however, not significant (*p* = 0.1299).

Furthermore, we ran a meta-regression to test the effect of preconception. Fifteen effect sizes did not include preconception cases in their sample, whereas 27 effect sizes included children exposed to a disaster in the preconception period. There was a significant positive overall correlation between PNMS levels and physical outcomes in preconception-included effect sizes (*r* = 0.1316; SE = 0.0293; *p* < 0.0001) (Figure 13), while the overall correlation was not significant in the no-preconception-included effect sizes (*r* = 0.0597; SE = 0.0374; *p* = 0.1105). Including preconception cases in the sample or not did not make a significant difference in the overall association between PNMS and physical development (*p* = 0.1299).

Next, we tested the effect of child age on the association between PNMS and physical development. We retrieved 24 effect sizes in children under 10 years of age and 18 in children aged 10 years or older. There was a significant positive overall correlation between PNMS and physical development in both children under 10 years of age (*r* = 0.0715; SE = 0.0286; *p* = 0.0124) and children aged 10 years or older (*r* = 0.1600; SE = 0.0375; *p* < 0.0001) (Figure 13). The comparison of the summary effects between the two age categories revealed no significant difference (*p* = 0.0604).

Finally, we could not run meta-regressions to test the type of report effect since all outcomes were reported by a third-party observer.

The Fisher’s test showed that there was a significant association between the child age and the timing of exposure effects (*p* < 0.0001): effects sizes collected in children aged 10 or older all included preconception cases while most effect sizes collected in children under the age of 10 did not include preconception cases. There was also a significant association between the age and the type of disaster effects (*p* < 0.0001): effect sizes extracted from flood studies were all collected in children aged under 10, while most effect sizes extracted from ice storm studies were collected in children aged 10 or older. The timing of exposure and the type of disaster effects were also significantly associated (*p* < 0.0001): all effect sizes extracted from ice storm studies included preconception cases while none of the effect sizes extracted from flood studies did. We were therefore unable to distinguish the effect of these factors from one another.

#### 3.3.5. Socio-Emotional Outcomes

This category of outcomes included: temperament (fussy/difficult, unadaptable, dull, needs attention, negative reactivity, shy-inhibition (approach-withdrawal), attentional control (persistence), approach, rhythmicity, cooperation-manageability, activity-reactivity, irritability) and socio-emotional functioning (socio-emotional problems and competence, personal-social skills). A correlation above zero suggested that greater PNMS levels were associated with worse socio-emotional outcomes. A total of 57 effect sizes extracted from 5 different studies were included in this analysis (see Figure 14). Overall, there was a significant positive association between PNMS and socio-emotional outcomes (*r* = 0.0588 (95% CI = [0.0304; 0.0871]; Z = 4.06; *p* < 0.0001)), such that greater PNMS levels were associated with worse socio-emotional outcomes. The sensitivity analysis revealed that the overall effect remained significant no matter which effect size was removed. We found low heterogeneity between the studies (I^2^ = 25.4% [0.0%; 46.6%]). We performed the trim and fill procedure [29] and found that no supplemental study would be required to adjust for publication bias (Figure 15).

We performed a meta-regression to determine the effect of the type of PNMS measures on the socio-emotional outcomes: objective hardship (15 effect sizes); psychological distress (33 effect sizes); cognitive appraisal (9 effect sizes); and diet change (0 effect size). There was a significant positive overall correlation between the socio-emotional outcomes and both objective hardship (*r* = 0.0679; SE = 0.0281; *p* = 0.0156) and psychological distress (*r* = 0.0681; SE = 0.0191; *p* = 0.0004) such that both objective hardship and psychological distress levels were associated with worse socio-emotional outcomes (Figure 16). The overall correlation between cognitive appraisal and socio-emotional outcomes was, however, not significant (*r* = 0.0091; SE = 0.0367; *p* = 0.8053). The summary effect of the three types of PNMS measures on the socio-emotional outcomes did not significantly differ (objective hardship vs. psychological distress (*p* = 0.9968); objective hardship vs. cognitive appraisal (*p* = 0.2027); psychological distress vs. cognitive appraisal (*p* = 0.1537)).

Next, we tested the effect of the type of natural disaster on the relationship between PNMS levels and socio-emotional outcomes. A total of 49 effect sizes were related to a flood, while eight were related to an ice storm, and none were related to an earthquake. There was a significant positive overall correlation between PNMS and socio-emotional outcomes in both flood related effect sizes (*r* = 0.0467; SE = 0.0153; *p* = 0.0022) and ice storm related effect sizes (*r* = 0.1299; SE = 0.0368; *p* = 0.0004) (Figure 16), and the overall correlation found in the ice storm effect sizes was significantly higher than in the flood effect sizes (*p* = 0.0370).

We could not run meta-regressions to test the type of report effect (only maternal reports), the timing of exposure effect (no preconception-included effect sizes), nor the age effect (all effect sizes in children under 10 years of age).

According to the Fisher’s test, the factors tested in the meta-regressions were not significantly dependant, meaning that the effect found with a given factor does not seem to be attributable to any of the other factors that we tested.

#### 3.3.6. Behavioral Outcomes

This category of outcomes included the following: internalizing (e.g., anxiety, depression) and externalizing (e.g., aggression) behaviors, attention, sleep problems, and autistic-like symptoms or traits. A positive correlation meant that greater PNMS levels were associated with worse behavioral outcomes. We retrieved 46 effect sizes extracted from six studies (Figure 17). Their combination resulted in a significant positive association between PNMS levels and behavioral outcomes in children (*r* = 0.0959 (95% CI = [0.0606; 0.1310]; Z = 5.30; *p* < 0.0001)), such that greater PNMS levels were associated with worse behavioral development in children. The sensitivity analysis suggested that no specific effect size included in the analysis was pulling the correlation. According to the trim and fill procedure [29] we performed, no supplemental study would be required to adjust for publication bias (Figure 18). We found low heterogeneity between the studies (I^2^ = 34.6% [6.0%; 54.5%]).

We ran meta-regressions to determine the effect of the type of PNMS measures on behavioral development: objective hardship (14 effect sizes); psychological distress (22 effect sizes); cognitive appraisal (10 effect sizes); and diet change (0 effect size). There was a significantly positive overall correlation between behavioral outcomes and both objective hardship (*r* = 0.1225; SE = 0.0327; *p* = 0.0002) and psychological distress (*r* = 0.1118; SE = 0.0259; *p* < 0.0001) such that higher objective hardship or psychological distress levels were associated with worse behavioral outcomes (Figure 19). However, the overall correlation between cognitive appraisal and behavioral outcomes was not significant (*r* = 0.0318; SE = 0.0364; *p* = 0.3816). The summary effect of the three types of PNMS measures on behavioral outcomes did not differ significantly (objective hardship vs. psychological distress (*p* = 0.7978); objective hardship vs. cognitive appraisal (*p* = 0.0640); psychological distress vs. cognitive appraisal (*p* = 0.0733)).

We then tested the effect of the type of natural disaster on the association between PNMS levels and behavioral outcomes using a meta-regression. There were 38 effect sizes related to a flood, 8 related to an ice storm, and none related to an earthquake. We found a significant positive overall correlation between PNMS and behavioral outcomes in both flood-related effect sizes (*r* = 0.0752; SE = 0.0170; *p* < 0.0001) and ice-storm-related effect sizes (*r* = 0.2643; SE = 0.0499; *p* < 0.0001) (Figure 19). The summary effect found in the ice storm group was significantly higher than in the flood group (*p* = 0.0003).

We tested the effect of the type of report on the relationship between PNMS and behavioral outcomes. We retrieved 41 effect sizes for which the outcome has been reported by the mother and 5 that have been reported by a third-party observer. No effect sizes were reported in medical reports. The association between PNMS levels and behavioral outcomes was significantly positive for maternal report (*r* = 0.1040; SE = 0.0190; *p* < 0.0001) but was not significant for the third-party observer report (*r* = 0.0236; SE = 0.0574; *p* = 0.6802) (Figure 19). The effect of the two types of report did not significantly differ (*p* = 0.1833).

Furthermore, we performed a meta-regression to test the effect of age on the association between PNMS and behavioral outcomes. This analysis included 40 effect sizes in children under 10 years of age and 6 in children aged 10 years or older. There was a significant positive overall correlation between PNMS and behavioral outcomes in both children under 10 years of age (*r* = 0.0923; SE = 0.0189; *p* < 0.0001) and aged 10 years or older (*r* = 0.1449; SE = 0.0668; *p* = 0.0300) (Figure 19). The comparison between the effect of the two age groups suggested no significant difference (*p* = 0.4485).

We could not run meta-regressions to test the timing of exposure effect because only two effect sizes included preconception cases.

According to Fisher’s test, there was a significant association between child age and type of disaster (*p* <0.0001). All effect sizes extracted from flood studies were collected in children aged under 10 while effect sizes extracted from ice storm studies were mostly collected in child aged 10 or older. It was therefore impossible for us to distinguish the effect of these factors from one another in the meta-regression in which they were involved.

## 4. Discussion

To our knowledge, this is the first meta-analytic review quantifying the effect of natural disaster-related PNMS on child/adolescent development from multiple studies. The main analyses revealed that PNMS has a significant effect on all spheres of development: birth outcomes, and cognitive, motor, physical, socio-emotional, and behavioral development. More precisely, we found that higher PNMS levels were associated with larger birth weights and head circumference and longer gestational age, and with greater BMI and total/central adiposity across childhood and adolescence. The surprising result with respect to birth outcomes is echoed in recent studies that have found that the COVID-19 pandemic decreased the incidence of preterm birth [69,70]. Higher PNMS was also consistently associated with worse cognitive, motor, socio-emotional, and behavioral outcomes. The secondary analyses uncovered the effect of different factors in these associations: type of PNMS (objective hardship, psychological distress, cognitive appraisal and diet change), type of natural disaster (ice storm, flood/cyclone or earthquake), type of report (medical report, maternal report, third-party observation), timing of exposure in pregnancy, and age of the child at assessment.

The type of PNMS had a significant effect on all the outcomes tested: birth outcomes and cognitive, motor, physical, socio-emotional and behavioral outcomes (Table 2). Measures of objective stress had a significant effect on all measured outcomes. We found a significant effect of psychological distress on all child development outcomes except for the birth outcomes for which we could not test the meta-regression with this category of PNMS due to too few cases. For the types of outcomes for which we were able to include a cognitive appraisal in our meta-regression analyses (cognitive, motor, physical, socio-emotional and behavioral), we found no significant relationship between this type of PNMS and the development of the child. Recent reports [55,61,64,71] suggest that the mother’s cognitive appraisal of the disaster has a stronger moderating effect than main effect, which would be consistent with the stress model of Lazarus and Folkman [18]. The moderating effect of cognitive appraisal has been reported in studies of maternal mental health during COVID [72,73]. For diet change, we were only able to test the effect with one type of outcome (birth outcomes) and that effect was significant.

The type of natural disaster had a significant effect on all outcomes except for the birth outcomes in which the meta-regression could not be run. Although both ice storm and flood effects were significant, the ice storm effect was significantly larger than that observed for the floods on all outcomes tested, except for physical outcomes. This difference might be explained in terms of differences in the characteristics of these disasters. First, ice storms are exclusively cold-weather/winter disasters whereas floods typically occur during warmer conditions (e.g., spring, summer, tropical climates). In terms of the studies included in this review, the 1998 Quebec ice storm resulted in loss of electricity for up to 44 days for the participating women during the coldest months of the year. As such, these women, in addition to the many hardships that were measured, could also have been exposed to extreme or sustained cold (not measured), which might have impacted their unborn children to a greater degree than for women who were exposed to power loss during summer in the two flood studies (2008 Iowa Flood Study and the 2011 Queensland Flood Study) which accounted for most of the flood-related outcomes reported in this review. Secondly, the 1998 Quebec ice storm impacted a larger geographical area than did the two floods from which most of the flood-related outcomes were obtained. As such, the 1998 Quebec ice storm was much more likely to have caused greater daily hardships (electrical loss, loss of income, damage to residents, daily threats to wellbeing) for a larger percentage of the general population than did the floods, for which the effects were limited to fairly circumscribed geographical areas. Unfortunately, we were unable to include the earthquake studies in these analyses, for which large geographical areas are also affected, to confirm if the magnitude of the disaster on the population differentially effects the objective hardship levels of pregnant women.

For cognitive, motor, and behavioral child outcomes, results using at least two types of reports were available, so the effect of type of report on the association between PNMS and these outcomes could be tested. However, the effect of type of report could not be tested for birth, physical, and socio-emotional outcomes, since all birth outcomes were medically reported, all physical outcomes were reported by a third-party observer, and all socio-emotional outcomes were reported by the mother. For future research, it would be interesting to ask more than one observer to cross-check the observations and ensure better reliability of the results, but also to check whether differences in perception can influence the results. Significant associations were observed between PNMS and cognitive, motor, and behavioral child outcomes reported by a third-party observer. For maternal reports, significant relationships between the PNMS and the behavioral and motor child outcomes were observed, but not with the cognitive outcomes. It is possible that mothers underestimate the cognitive difficulties of their children, such as their productive and receptive language. It is also possible that mothers adapt to their children’s mode of communication, which could make it difficult for them to take a step back and assess their children’s language abilities relative to their peers. Finally, although one would have thought that the type of report might have made a difference in the magnitude of the association between PNMS and child development, our results do not suggest that this makes a significant difference.

For the outcomes where the effects of timing of exposure could be assessed (birth and physical outcomes), we found that for studies that included preconception-exposed children, PNMS was associated with less-than-optimal birth and physical outcomes. When studies were restricted to in utero-exposed children only, PNMS was associated only with worse birth outcomes. This suggests that birth outcomes (which can be seen as proxy measures of fetal growth, particularly birth weight and length, and ponderal index) can be influenced by alterations in the uterine environment at any time during pregnancy, while later physical development might require that the uterine environment be altered prior to conception and implantation. However, this speculative interpretation warrants further investigation.

We observed a significant association between PNMS and child development in physical and behavioral outcomes in the child aged under 10 years, but also aged 10 or older, however we did not detect a significant difference between the two age categories. We could not test this effect in the other child outcomes. Still, available data suggest that the effects of PNMS on behavioral and physical difficulties in childhood also persist into adolescence, highlighting the importance of early prevention and/or treatment programs.

The results of Fisher’s tests indicate, however, that some effects still must be distinguished from each other due to the design of the studies that we included in our meta-analyses. For both the cognitive and the motor outcomes, we found that the effects of both type of report and the type of disaster were significantly associated. For the cognitive outcomes, most effect sizes from the ice storm study were for outcomes reported by a third-party observer. The association was stronger in the motor outcomes since all the effect sizes extracted from the 1998 ice storm study were for outcomes reported by third-party observers, while effect sizes extracted from flood studies were mostly for outcomes reported by mothers. For both the physical and the behavioral outcomes, we found a significant association between child age and the type of disaster such that all the effect sizes extracted from the flood studies were for outcomes assessed in children under age 10 while most of the effect sizes extracted from the 1998 Quebec Ice Storm Study were for outcomes assessed in children aged 10 or older. For the physical outcomes only, we also found that all the effect sizes extracted from the 1998 Quebec Ice Storm Study included preconception cases in their samples, while the effect sizes extracted from flood studies did not. Additionally, the distribution of child age categories was not independent of the distribution of timing of exposure and type of natural disaster categories. These patterns made it impossible to distinguish the effect of the type of natural disaster, the timing of exposure and the child age. To summarize, the relative importance of the predictors of the effect sizes in this review is difficult to determine given the confounding nature of those variables.

Many cases of mediations and moderations are not reflected in this meta-analytic review. For example, among the studies included, some tested interactions between PNMS and sex of the child, between PNMS and timing of exposure, or between the PNMS measures [47]. Others tested mediations through other PNMS measures (e.g., diet change [39]) or predictive characteristics of the child (e.g., birth weight [42]). Meta-analyses that include findings from mediation and moderations analyses are required to better understand the effects of disaster-related PNMS on child development. Thus, the magnitude of the influence of disaster-related PNMS might be higher than that observed in the present study in which only PNMS main-effects were studied.

Quality assessment of studies that are not randomized controlled trials is still challenging since the tools available to assess the risk of bias in observational studies of exposures are not yet well developed [28]. There is also clearly an issue concerning the tools currently used in the literature to measure some types of PNMS. There are published reliability and validity data for all instruments used to measure psychological distress (e.g., PTSD Checklist Civilian Version (PCL-C) [74]; PTSD-like Symptoms (Impact of Event Scale-Revised (IES-R)) [75]; peritraumatic distress (Peritraumatic Distress Inventory (PDI)) [76]; and peritraumatic dissociation (Peritraumatic Dissociative Experiences Questionnaire (PDEQ)) [77]. In some cases, a composite score or a translation/adaptation of these scales was used [59]. However, none of the instruments used to measure objective hardship (e.g., Storm32; Iowa Flood 100 (IF100); Queensland Flood Objective Stress Scale (QFOSS); Financial, Evacuation and Physical Strain Indexes) or cognitive appraisal have been validated by conventional means. One instrument related to diet change was based on a previously validated measure [59], but none of the other instruments related to food and water intake have psychometric data. The non-validated instruments were developed by the study authors to assess specific characteristics of the catastrophic situation experienced by the participants. Sharing these instruments could facilitate their validation with various populations.

### 4.1. Limitations

First, this meta-analytic review only included studies in which disaster-related PNMS could be directly linked to pregnant women themselves (i.e., direct measures of PNMS reported by individual pregnant women were required for inclusion). As such, we excluded the large literature comprising many large-scale epidemiological or population-based studies assessing the effects of natural disasters on child outcomes, in particular birth outcomes; for those studies, PNMS was assessed by including exposure or non-exposure categories according to whether pregnant women were believed to be residing within a specific geographical area at the time of the disaster. Even though individual-level assessments of disaster-related PNMS could be more reliable, the sample characteristics can be problematic; the cohorts of the included studies are relatively small compared to population-based studies and they are comprised primarily of well-educated white females, which limits the generalizability of the results. Second, since this review only includes correlational studies, causal conclusions about the influence of PNMS on child development cannot be drawn, although the quasi-experimental design of disaster studies approaches true experiments and, thus, can support dose–response associations. Third, most studies included in this meta-analytic review were conducted in industrialized countries (Australia, Canada, and United States of America). This could explain the publication bias we detected in the funnel plots. The lack of resources faced by women in more disadvantaged regions of the world (e.g., Central America) makes them more vulnerable economically, socially, politically, physically, and psychologically [78]. It would, therefore, be important to determine whether disaster-related PNMS effects reported here can be replicated in low- and middle-income countries. Additionally, 25 of the 30 studies reported here come from the same research group (SPIRAL), which used very similar methods; this can be seen as a weakness in that this limits the independence of the investigations while, on the other hand, it can be seen as a strength by limiting the diversity in the methods used. Nonetheless, the sensitivity analysis revealed that none of the results we obtained were pulled by an artifact. We also performed the trim and fill procedure to check if the results would remain significant when accounting for this risk, and they have indeed remained significant in all cases. An additional limitation from the numerous effect sizes per study is the interdependence of the effect sizes included in each meta-analysis. Multiple approaches were explored to address this issue, including multivariate analysis [79], cross-classified random-effects models (CCREM) [80], and robust variance estimation [81]. However, we concluded that the data in our meta-analyses do not lend themselves to any of these approaches for a variety of reasons, the main one being the small number of individual projects. Since we could not account for the dependence of the effect sizes in each meta-analysis, the results of this review should be interpreted cautiously. Future findings in the field may allow for new methods that account for dependence among studies to be used. Finally, although the effect sizes reported in these meta-analyses often explains less than 3% of the variance in the outcomes, and may not seem “clinically meaningful” at an individual level, small increases in the prevalence of chronic health problems, such as obesity, in a population could have a large economic burden on that population and should be considered by public health agencies [82].

### 4.2. Strengths

This meta-analytic review is unique in making the distinctions between different categories of PNMS, namely, objective hardship, psychological distress, cognitive appraisal, and diet change. PNMS was defined to ensure that the women included in the analyses actually experienced the disaster. The timing of exposure was also defined in a way that made it possible to circumscribe the effect of preconception cases in the association between PNMS and child development. Additionally, since the SPIRAL studies used identical or very similar methods and measurement instruments, the variation among the methods is reduced, and comparisons between PNMS types are more valid. Finally, this meta-analytic review included a total of 296 effect sizes extracted from 30 studies (seven different natural disasters).

## 5. Conclusions

The results of this meta-analytic review demonstrate that natural disaster-related PNMS significantly influences child development in multiple spheres: birth outcomes and cognitive, motor, physical, socio-emotional, and behavioral development. PNMS is often a catch-all concept. This study disentangles the effects of multiple factors that could be confused in their relative roles in the association between PNMS and child development. It is now clear that the type of PNMS and the type of natural disaster influence this association. For the type of PNMS, effect sizes for objective hardship were almost consistently the highest, despite low absolute magnitude. More severe psychological distress, but not a negative cognitive appraisal of the disasters, was also an important predictor of subsequent child outcomes. Diet change was also found to be an important predictor of birth outcomes. As for the type of natural disaster, the PNMS effect on child development was often greater for the sample exposed to an ice storm than for those exposed to floods. As such, disasters of greater duration and wider distribution in the population may be those for which first responder and public health agencies should apply greater protections for pregnant women and their unborn children in order to reduce downstream challenges to children’s health, development, and wellbeing.

## Figures and Tables

**Figure 1 ijerph-18-08332-f001:**
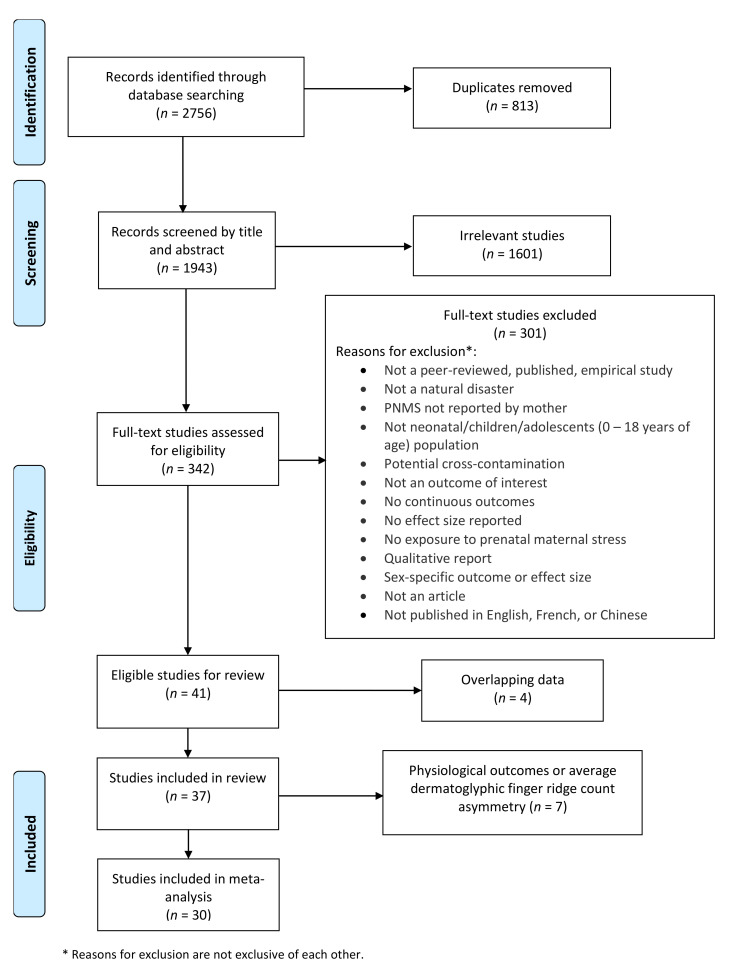
Flow chart.

**Figure 2 ijerph-18-08332-f002:**
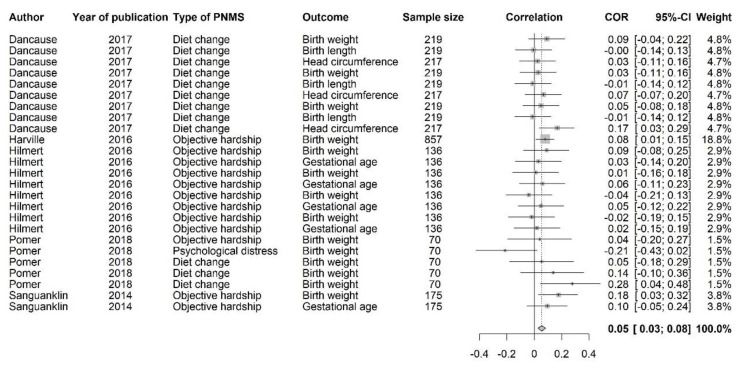
Birth Outcomes Forest Plot.

**Figure 3 ijerph-18-08332-f003:**
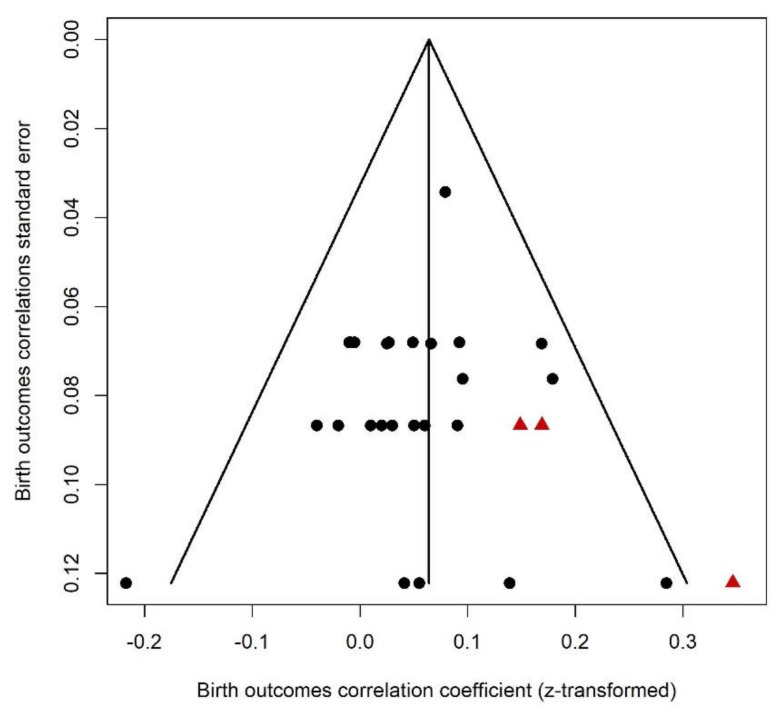
Birth Outcomes Funnel Plot. Three positive effect sizes were added and represented by red triangles.

**Figure 4 ijerph-18-08332-f004:**
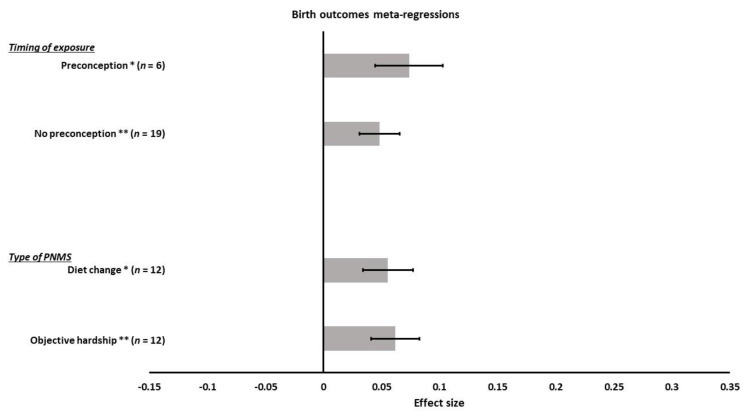
Birth outcomes meta-regressions. Type of PNMS effect (diet change; objective hardship) on birth outcomes and timing of exposure effect (no preconception; preconception) on the association between PNMS levels and birth outcomes. Note: * *p* < 0.05; ** *p* < 0.01.

**Figure 5 ijerph-18-08332-f005:**
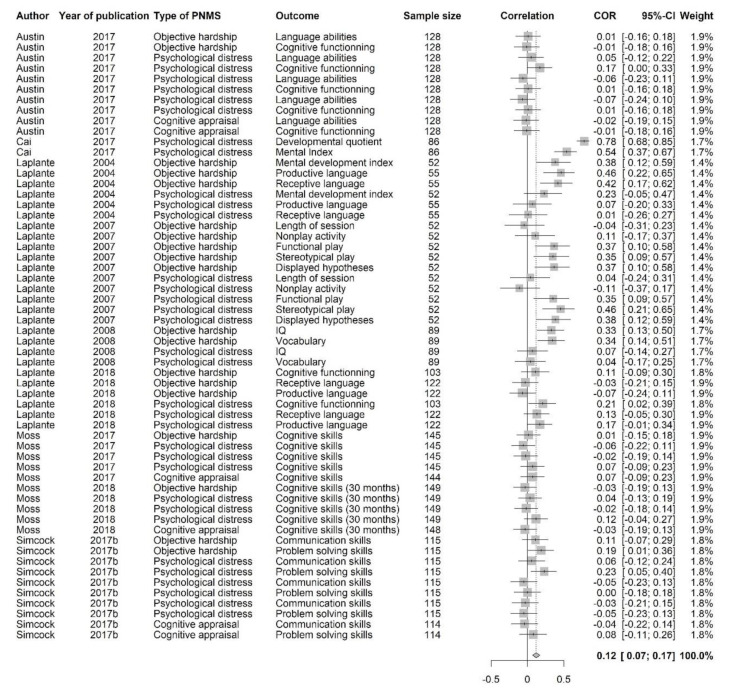
Cognitive outcomes forest plot.

**Figure 6 ijerph-18-08332-f006:**
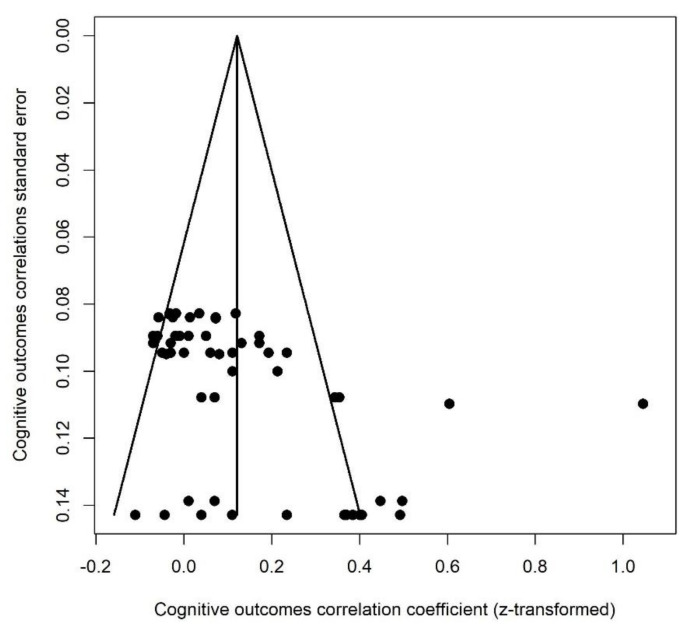
Cognitive outcomes funnel plot.

**Figure 7 ijerph-18-08332-f007:**
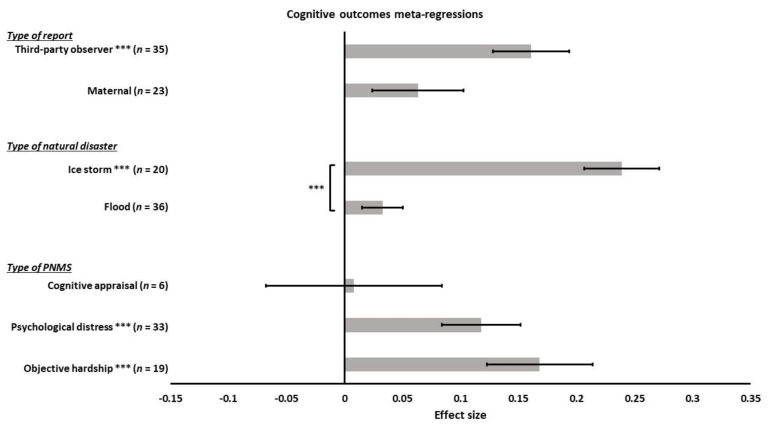
Cognitive outcomes meta-regressions. Type of PNMS effect (objective hardship; psychological distress; cognitive appraisal) on cognitive outcomes, type of natural disaster (flood; ice storm), and type of report (maternal; third-party observer) on the association between PNMS levels and cognitive outcomes. Note: *** *p* < 0.001.

**Figure 8 ijerph-18-08332-f008:**
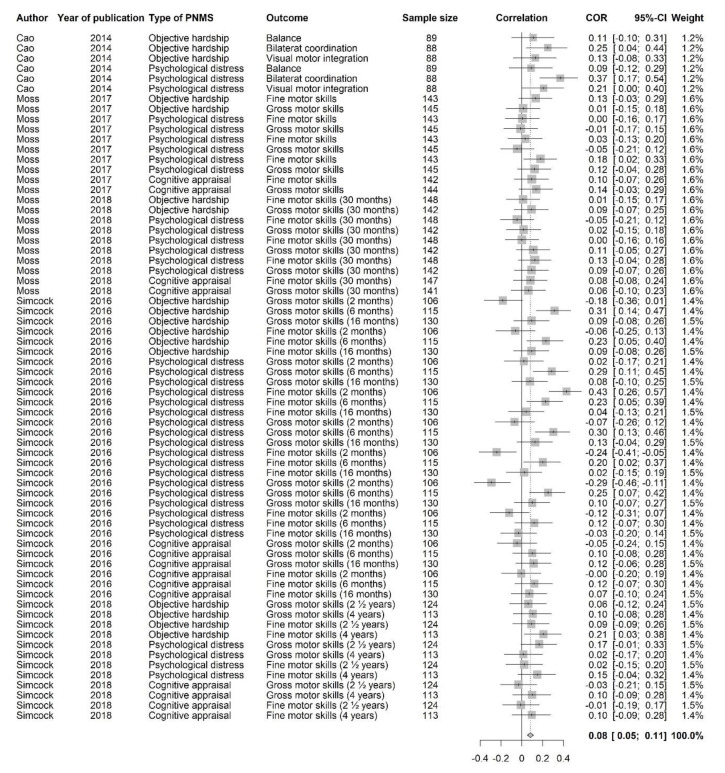
Motor outcomes forest plot.

**Figure 9 ijerph-18-08332-f009:**
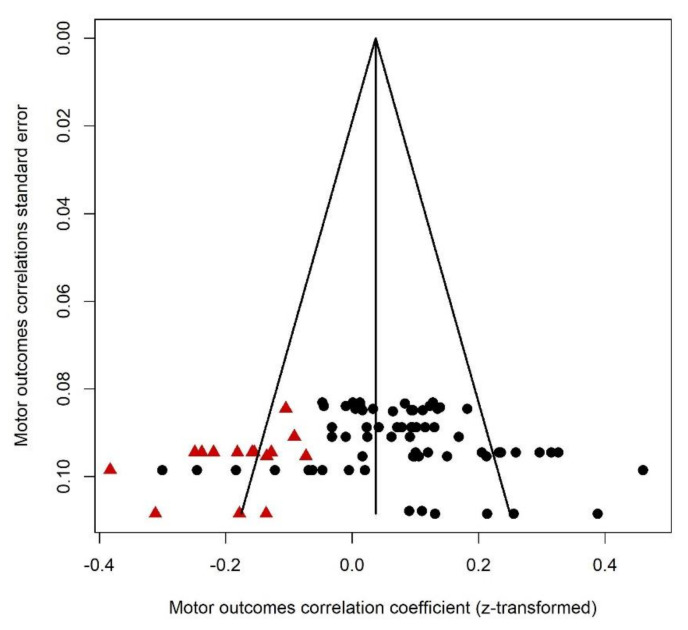
Motor outcomes funnel plot. Fifteen positive effect sizes were added and represented by red triangles.

**Figure 10 ijerph-18-08332-f010:**
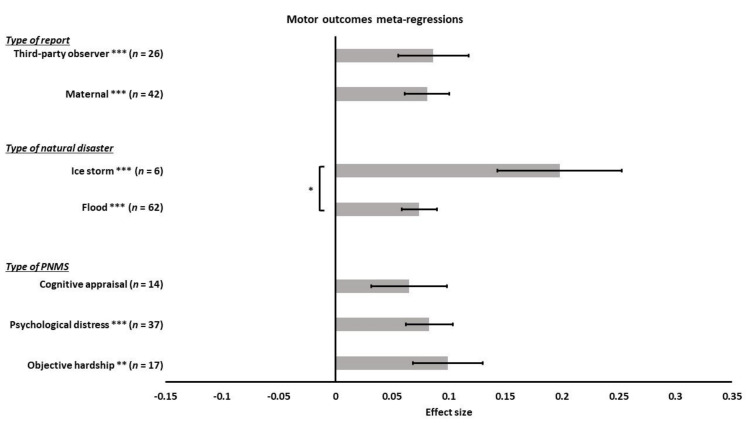
Motor outcomes meta-regressions. Type of PNMS effect (objective hardship; psychological distress; cognitive appraisal) on motor outcomes, type of natural disaster effect (flood; ice storm) and type of report (maternal; third-party observer) on the association between PNMS levels and motor outcomes. Note: * *p* < 0.05; ** *p* < 0.01; *** *p* < 0.001.

**Figure 11 ijerph-18-08332-f011:**
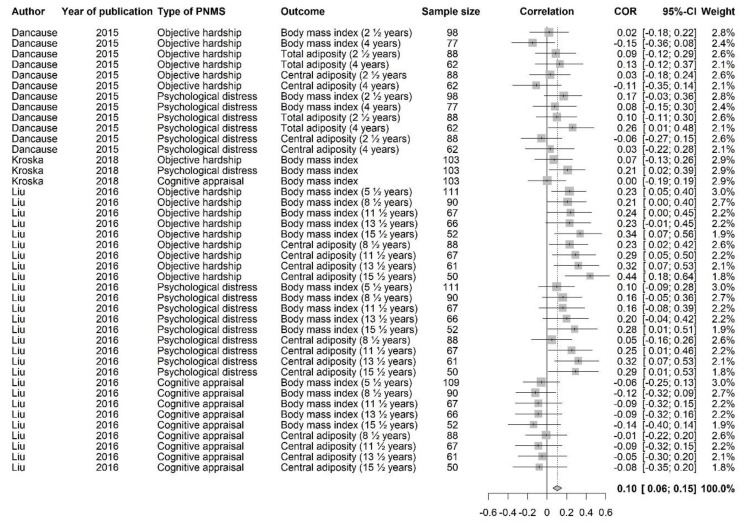
Physical outcomes forest plot.

**Figure 12 ijerph-18-08332-f012:**
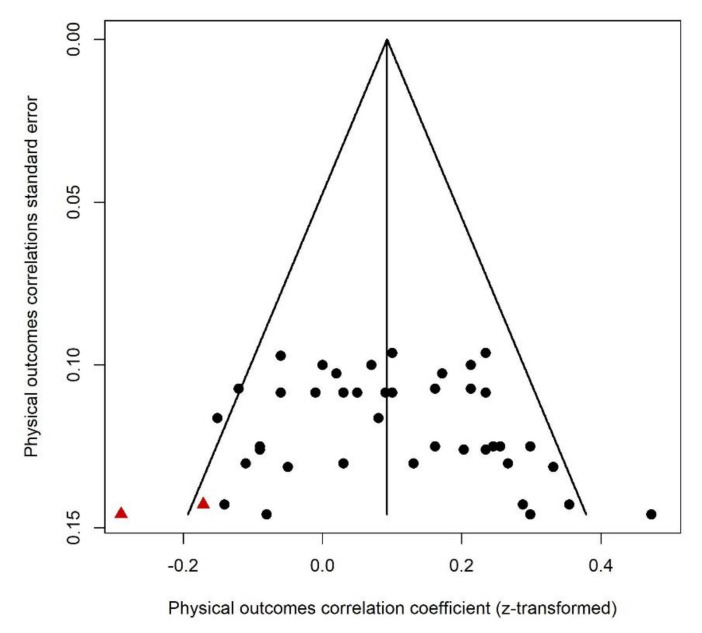
Physical outcomes funnel plot. Two negative effect sizes were added and are represented by red triangles.

**Figure 13 ijerph-18-08332-f013:**
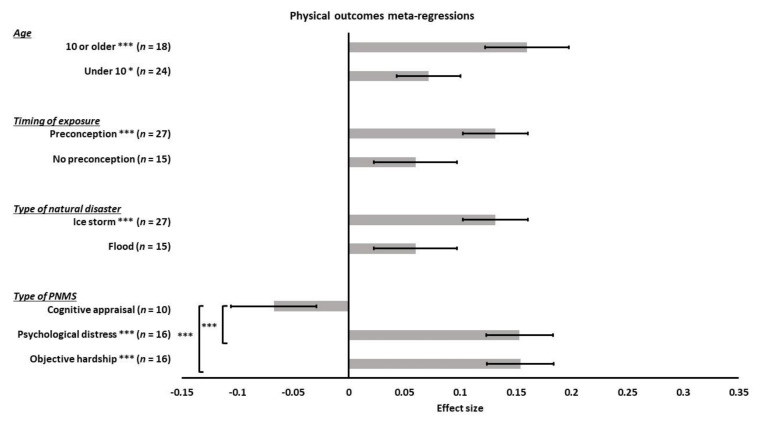
Physical outcomes meta-regressions. Type of PNMS effect (objective hardship; psychological distress; cognitive appraisal) on physical outcomes, type of natural disaster effect (flood; ice storm), timing of exposure effect (no preconception; preconception) and age effect (under 10 or 10 or older) on the association between PNMS levels and physical outcomes. Note: * *p* < 0.05; *** *p* < 0.001.

**Figure 14 ijerph-18-08332-f014:**
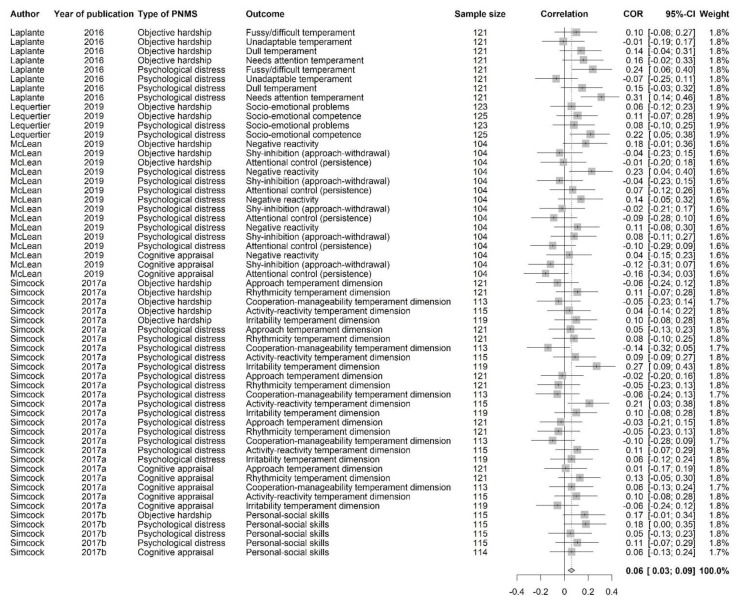
Socio-emotional outcomes forest plot.

**Figure 15 ijerph-18-08332-f015:**
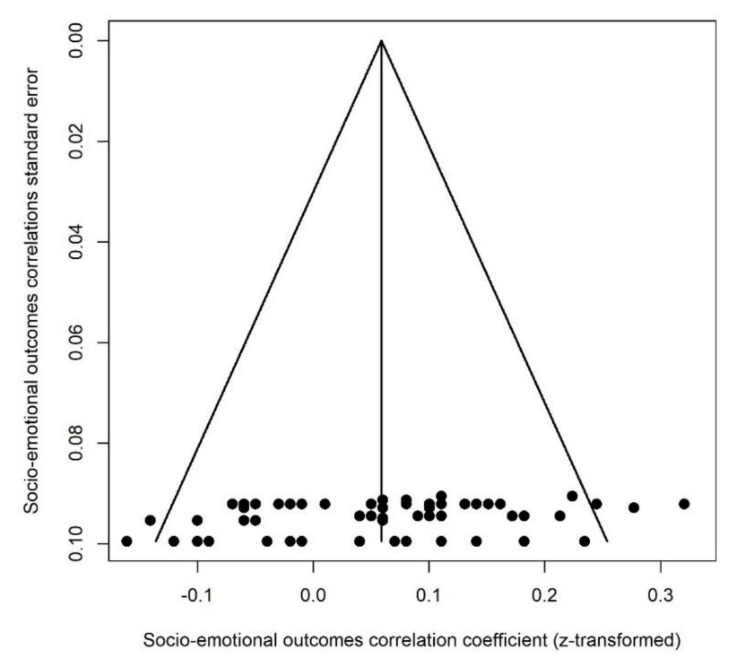
Socio-emotional outcomes funnel plot.

**Figure 16 ijerph-18-08332-f016:**
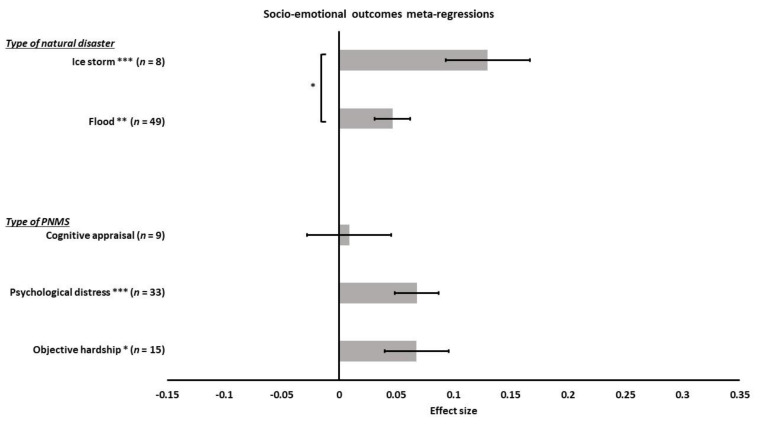
Socio-emotional outcomes meta-regressions. Type of PNMS effect (objective hardship; psychological distress; cognitive appraisal) on socio-emotional outcomes and type of natural disaster (flood; ice storm) on the association between PNMS levels and physical outcomes. Note: * *p* < 0.05; ** *p* < 0.01; *** *p* < 0.001.

**Figure 17 ijerph-18-08332-f017:**
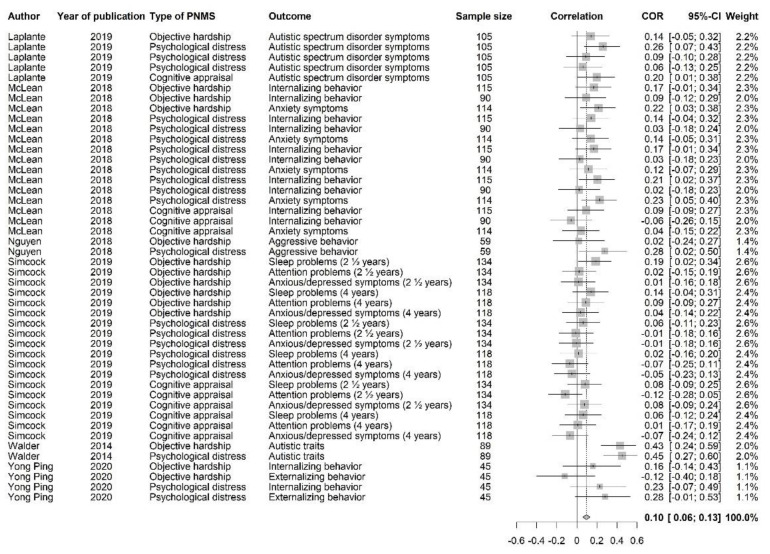
Behavioral outcomes forest plot.

**Figure 18 ijerph-18-08332-f018:**
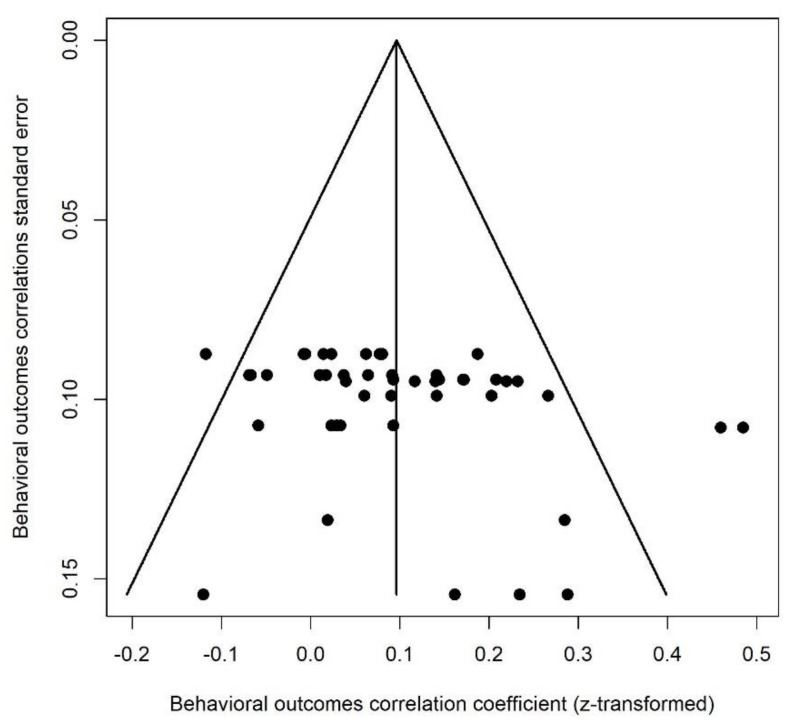
Behavioral outcomes funnel plot.

**Figure 19 ijerph-18-08332-f019:**
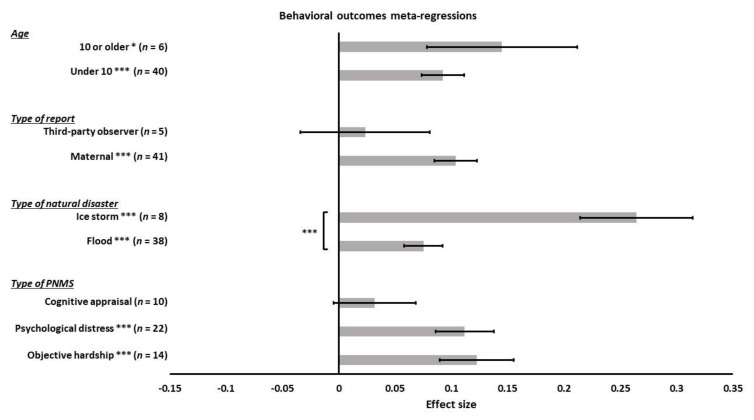
Behavioral outcomes meta-regressions. Type of PNMS effect (objective hardship; psychological distress; cognitive appraisal) on behavioral outcomes, type of natural disaster effect (flood; ice storm), type of report effect (maternal; third-party observer), and age effect (under 10 or 10 or older) on the association between PNMS levels and physical outcomes. Note: * *p* < 0.05; *** *p* < 0.001.

**Table 1 ijerph-18-08332-t001:** Characteristics of studies included in the review.

Author	Year of Publication	Country	Type of Outcomes (Specific Outcome [Type of Report])	Natural Disaster	Type of PNMS	Timing of Exposure	Age of Assessment	Sample Size
Austin	2017	Australia	Cognitive (language abilities [maternal], cognitive functioning [third-party observer])	2011 Queensland Flood	Objective; psychological; cognitive appraisal	No preconception	30 months	131
Cai	2017	China	Cognitive (developmental quotient, mental index [third-party observer])	2008 Sichuan Earthquake	Psychological	Preconception included (3 years)	0–4 years	86
Cao	2014	Canada	Motor (balance, bilateral coordination, visual motor integration [third-party observer])	1998 Quebec Ice Storm	Objective; psychological	No preconception	5 ½ years	89
Cao-Lei	2018	Canada	Physiological (c-peptide [medical])	1998 Quebec Ice Storm	Objective; cognitive appraisal	Preconception included (3 months)	13 ½ years	30
Dancause	2013	Canada	Physiological (insulin [medical])	1998 Quebec Ice Storm	Objective; psychological	Preconception included (1 month)	13 ½ years	32
Dancause	2015	United States	Physical (body mass index, total adiposity, central adiposity [third-party observer])	2008 Iowa Flood	Objective; psychological	No preconception	2 ½ and 4 years	106
Dancause	2017	Australia	Birth (birth weight, birth length, head circumference, head circumference to birth length ratio, ponderal index [medical])	2011 Queensland Flood	Diet	No preconception	Birth	222
Harville	2016	Haiti	Birth (birth weight [medical])	2010 Haiti Earthquake	Objective	Preconception included (20 ½ months)	Birth	857
Hilmert	2016	United States	Birth (birth weight, gestational age [medical])	2009 Red River Flood	Objective	No preconception	Birth	136
King	2009	Canada	Physical (dermatoglyphic asymmetry [third-party observer])	1998 Quebec Ice Storm	Objective; psychological; maternal cortisol	Preconception included (3 months)	4; 5; 5 ½ years	97
Kroska	2018	United States	Physical (body mass index [third-party observer])	2008 Iowa Flood	Objective; psychological; cognitive appraisal	No preconception	30 months	103
Laplante	2004	Canada	Cognitive (mental development index [third-party observer], productive and receptive language [maternal])	1998 Quebec Ice Storm	Objective; psychological	No preconception	2 years	58
Laplante	2007	Canada	Cognitive (length of session, nonplay activity, functional play, stereotypical play, displayed hypotheses [third-party observer])	1998 Quebec Ice Storm	Objective; psychological	No preconception	2 years	52
Laplante	2008	Canada	Cognitive (IQ, vocabulary [third-party observer])	1998 Quebec Ice Storm	Objective; psychological	No preconception	5 ½ years	89
Laplante	2016	Canada	Socio-emotional (temperament: fussy/difficult, unadaptable, dull, needs attention [third-party observer])	1998 Quebec Ice Storm	Objective; psychological	No preconception	6 months	121
Laplante	2018	United States	Cognitive (cognitive functioning [third-party observer], productive and receptive language [maternal])	2008 Iowa Flood	Objective; psychological	No preconception	30 months	132
Laplante	2019	Australia	Behavioral (autistic spectrum disorder symptoms [maternal])	2011 Queensland Flood	Objective; psychological; cognitive appraisal	No preconception	30 months	105
Lequertier	2019	Australia	Socio-emotional (socio-emotional problems and competence [maternal])	2011 Queensland Flood	Objective; psychological	No preconception	16 months	125
Liu	2016	Canada	Physical (body mass index, central adiposity [third-party observer])	1998 Quebec Ice Storm	Objective; psychological; cognitive appraisal	Preconception included (3 months)	5 ½; 8 ½; 11 ½; 13 ½; 15 ½ years	111
McLean	2018	Australia	Behavioral (internalizing behavior [maternal and third-party observer], anxiety symptoms [maternal]	2011 Queensland Flood	Objective; psychological; cognitive appraisal	No preconception	4 years	115
McLean	2019	Australia	Socio-emotional (temperament: negative reactivity, shy-inhibition (approach-withdrawal), attentional control (persistence) [maternal])	2011 Queensland Flood	Objective; psychological; cognitive appraisal	No preconception	16 months	104
McLean	2020	Australia	Physiological (cortisol [medical])	2011 Queensland Flood	Objective; psychological; cognitive appraisal	No preconception	16 months	111
Moss	2017	Australia	Cognitive (cognitive development); motor (fine and gross motor development [third-party observer])	2011 Queensland Flood	Objective; psychological; cognitive appraisal	No preconception	16 months	145
Moss	2018	Australia	Cognitive (cognitive development [third-party observer]); motor (fine and gross motor development [third-party observer])	2011 Queensland Flood	Objective; psychological; cognitive appraisal	No preconception	30 months	150
Nguyen	2018	Canada	Physiological (testosterone [medical], cortisol [medical]; behavioral (aggressive behavior [maternal])	1998 Quebec Ice Storm	Objective; psychological	Preconception included (3 months)	11 ½ years	59
Pomer	2018	Vanuatu	Birth (birth weight [medical])	2015 Cyclone Pam	Objective; psychological; diet	Preconception included (3 months)	Birth	70
Sanguanklin	2014	Thailand	Birth (gestational age [medical])	2011 Thailand Flood	Objective	No preconception (third trimester only)	Birth	175
Simcock	2016	Australia	Motor (fine and gross motor development [maternal])	2011 Queensland Flood	Objective; psychological; cognitive appraisal	No preconception	2 months; 6 months; 16 months	2 months (106); 6 months (115); 16 months (130)
Simcock	2017a	Australia	Socio-emotional (personal-social skills [maternal]); cognitive (communication; problem-solving skills [maternal])	2011 Queensland Flood	Objective; psychological; cognitive appraisal	No preconception	6 months	115
Simcock	2017b	Australia	Socio-emotional (temperament: approach, rhythmicity, cooperation-manageability, activity-reactivity, irritability [maternal])	2011 Queensland Flood	Objective; psychological; cognitive appraisal	No preconception	6 months	121
Simcock	2018	Australia	Motor (fine and gross motor development [maternal])	2011 Queensland Flood	Objective; psychological; cognitive appraisal	No preconception	2 ½ and 4 years	2 ½ years (124) and 4 years (113)
Simcock	2019	Australia	Behavioral (sleep problems, attention problems, anxious/depressed symptoms [maternal])	2011 Queensland Flood	Objective; psychological; cognitive appraisal	No preconception	2 ½ and 4 years	2 ½ (134); 4 (118)
Strahm	2020	United States	Physiological (cortisol [medical])	2009 Red River Flood	Objective; maternal cortisol	No preconception	9 years	56
Veru	2015	Canada	Physiological (lymphocytes, cytokines [medical])	1998 Quebec Ice Storm	Objective; psychological	Preconception included (3 months)	13 years	37
Walder	2014	Canada	Behavioral (autistic traits [maternal])	1998 Quebec Ice Storm	Objective; psychological	No preconception	6 ½ years	89
Yong Ping	2015	United States	Physiological (cortisol [medical])	2008 Iowa Flood	Objective; psychological	No preconception	2 ½ years	94
Yong Ping	2020	Canada	Behavioral (internalizing and externalizing behavior [maternal]); physiological (cortisol [medical])	1998 Quebec Ice Storm	Objective; psychological	No preconception	13 years	45

**Table 2 ijerph-18-08332-t002:** Effects of factors on the association between PNMS and child development outcomes. Numbers represent the PNMS effect for the factor category. Grey: not tested; Green: not significant; Yellow *: *p* < 0.05; Orange **: *p* < 0.01; Red ***: *p* < 0.001.

Factors/Type of Outcomes	Birth	Cognitive	Motor	Physical	Socio-Emotional	Behavioral
*Type of PNMS*						
Objective hardship	0.0618 **	0.1682 ***	0.0994 **	0.1539 ***	0.0679 *	0.1225 ***
Psychological distress		0.1178 ***	0.0827 ***	0.1532 ***	0.0681 ***	0.1118 ***
Cognitive appraisal		0.0082	0.0650	−0.0674	0.0091	0.0318
Diet	0.0555 *					
*Type of natural disaster*						
Ice storm		0.2389 ***	0.1978 ***	0.1316 ***	0.1299 ***	0.2643 ***
Flood		0.0329	0.0741 ***	0.0597	0.0467 **	0.0752 ***
*Type of report*						
Maternal		0.0633	0.0809 ***			0.1040 ***
Third-party observer		0.1607 ***	0.0865 ***			0.0236
*Timing of exposure*						
No preconception	0.0480 **			0.0597		
With preconception	0.0737 *			0.1316 ***		
*Age*						
Under 10				0.0715 *		0.0923 ***
10 or older				0.1600 ***		0.1449 *

## Data Availability

No new data were created or analyzed in this study. Data sharing is not applicable to this article.

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
