# Peer review of "Effect of Natural Disaster-Related Prenatal Maternal Stress on Child Development and Health: A Meta-Analytic Review"

_ijerph, 2021, doi:10.3390/ijerph18168332_

Round 1
Reviewer 1 Report
Very interesting work with great applicability. This manuscript explores how women's exposure to natural disasters during pregnancy can affect their offspring in important aspects such as anthropometric changes, motor development or cognitive skills. The overall idea is very well summarized in the first paragraph of the discussion. The article reports a wealth of data with a thorough analysis of them. However, I would like to suggest some comments to improve the quality of the article and its interpretation.
- My main concern is the publication bias that can be observed in the funnel plots. This should be discussed and explained in more detail. In addition, the articles found have a lot of homogeneity, which could be linked to the publication bias. This could be due to the fact that the articles focus on industrialized countries and have not considered other such as those from Latin-America where the proportion of floods, hurricanes, or earthquakes is high (Argentina and Chile could be examples of this). This aspect need to be clarify in the limitation section.
- It should be explained as a correlation > 0 in the PNMS is interpreted as larger birth outcomes while in the case of cognitive variables or motor skills it would be interpreted as worse results. I imagine that this is due to the way the measurements are taken in the original articles.
- I do not know if I lost it but I cannot found the final 37 articles selected in the reference list.
- The statistical section is very extensive and useful but at the same time it contains information that does not exactly belong to the statistical analysis (i.e. the relevance of cortisol onwards). I suggest to summarize it more and focus it on what are the applied statistical tests, the meta-regressions, the adjustment variables and the p-values considered as significant.
- Maybe, I may have missed it, but I did not see in the material and methods how motor or cognitive outcomes were defined. It might be interesting how, overall, the authors of the original articles define these concepts since "balance" (line 391), “temperament”, or “socio-emotional functioning” (line 527) for example, can be interpreted in different ways.
- In section 3.3.6. It is not clear at what point in pregnancy the articles variables are measured. This could be important in relation to sleep problems (line 574), since during pregnancy, melatonin is increasing and is a highly fluctuable hormone intra-individually. This hormone is one of the main inducers of sleep/wake cycles.
- The discussion is a very well structured and well written. My concerns are how could the authors explain the positive association between PNMS and gestational age (line 654-655). There is evidence that stress/anxiety during pregnancy is associated with prematurity (PMID: 21890014; PMID: 12505886; PMID: 26772181).
- On the other hand, I miss a biological explanation of how changes during pregnancy may affect fetal programming. Perhaps an adaptation to PNMS during pregnancy produces modification of hormonal patterns (such as cortisol), cytokines or inflammatory response (elevated leukocytes), which may be adaptive early on but interfere with fetal development. This links with the fetal programming and DOHaD model of induction. In long term, may be maladaptive for neonatal development. Kindly suggest review PMID: 29875698 and https://doi.org/10.1007/978-3-319-40007-5_32-1
Minor comments
- The results in the abstract need to say something more. Particularly, what PNMS was associated with what outcomes.
- I suggest summarizing the introduction and moving lines 35-37 to the end. The paragraph between lines 48-62 is too long. The most relevant for the article is between lines 63-77 and 90-97.
- On the other hand, the paragraph between lines 100-110 seems more of material and methods.
- Between lines 119-132 is a potpourri, the first part is already mentioned above, and the exclusion articles should move to both the material and methods and the limitations of the study.
- Section 2.2. What MeSH terminology were used?
- Lines 158-163. Why is relevant the epigenetics factors in the context of the article?
- Section 2.4. I suggest to modify subjective terminology (“good”) by low/moderate/high.
- The figure 1 needs an introductory text. In addition, footnote needs to clarify what means “with reasons”.
- The table 1 needs to clarify that articles were included because the final N was 37 and only appears 9.
- Figure 5. Could be grouped by “outcome” instead of by author? Its looks like not all cognitive outcome are affected the same way by PNMS. It could be interesting to discuss.
- I suggest to move section 3.4. into the discussion.
Reviewer 2 Report
This meta-analysis study did a extensive search for literature. However, there are some major shortcomings in the meta-analysis methods:
- Figure 1 didn't clearly state why some of the studies were excluded.
- For each of included studies, didn't evaluate the quality of each study at all, such as biasness, study design and others.
- For each outcome, whether random-effect or fixed effect model were used?
Some minor mistakes:
- It is hard to understand why authors grouped different outcomes and types of PNMS in one meta-analysis result. They should be seperated for analysis.
- On line 230, it is Fisher Z transformation and it is transform all correlation coefficients to normally distributed variables.
Reviewer 3 Report
My role as a reviewer here was to review the statistical methods used, and so I will need to leave the merits of the research hypotheses etc. to the other reviewers. But on my first read I was very concerned that correct methods have not been used, or at least that it is not clear to me that they are. So I will details this major concern below for consideration.
A standard meta-analysis that includes multiple effects from the same study is often misleading. A standard meta-analysis requires independence between the effects being pooled, but when the same study is used more than once then that assumption is broken. This can happen for various reasons. E.g. the same cohort is used in full, or even in part, to calculate more than one of the effects being meta-analyzed. Secondly, even independent groups within a single study can not be seen as independent in the context of a meta-analysis. That is being effects from even different groups in single study are more likely to be similar than effects from different studies (e.g. due to things like similarities of cohort etc.). An example recent discussion on this can be found in "A guide to conducting a meta-analysis with non-independent effects sizes" (2019) by Cheung. This is often not noticed as important at the time, but unfortunately encourages scrutiny of the work later. While a bit of this happening would perhaps be ignored since in the scheme of things the impact could be trivial, when it is happening a lot then questions need to be asked. E.g., the pooled estimates and conclusions from Figure 5, 8, 11, 14, 17 etc. and presumably the meta-regressions, would raise concerns.
So I think there are a few options here, I think. Firstly, the appropriate methods have been used to account for dependence but the details haven't been included. In that case they should be described. Secondly, dependence and therefore misleading findings are not of concern here. In that case it needs to be make very clear why that is the case. Finally, the analyses be re-done with methods accounting for dependence.
Other comments
Additionally, the pooled correlations are very small. Correlations of such a small magnitude would look random noise. Are these clinically meaningful? It is not uncommon for a meta-analysis with many pooled effects to find an extremely small effect which has no clinical meaning. So it must be made clear that they are.
More details on the meta-analysis methods are needed. E.g. fixed or random effect models? If random effects then which estimator of heterogeneity variance was used (i.e. DerSimonian & Laird?).
Reviewer 4 Report
In this article, Lafortune and colleagues performed a meta-analysis aimed to determine the effects resulting from maternal exposure to natural disasters, just before and during pregnancy, on the offspring.
The analysis is comprehensive and clearly described, and the diverse outcomes evaluated were assayed by different approaches to ensure the significance of the main findings.
Some minor issues may be addressed to further improve the manuscript:
In Introduction, references concerning violence against women should be included as a common factor of PNMS.
In Discussion, comments on the socio-economic diversity among the populations included in this study is required. For instance, some reports may include women that live under chronic social stress before and after the natural disaster, and this may impact on the effects studied.
Please specify if neonatal data was adjusted by gestational age.
Please fix the first bar in PNMS of figure 13
Reviewer 5 Report
Dear authors,
The observations will be listed below:
Line 148: It is not clear why the study approach is considered psychological and epidemiological not. I am not able to verify the justification for this claim.
Line 177: In the selection of studies, larger samples do not necessarily mean significant. I think this selection criteria should be better explored.
Line 277: Table 1 seems to me incomplete and incompatible with the selection made.
Line 300: Figure 2 could be converted into a Table, grouping by PNMS type. I don't see it as a figure. The CIs of the correlation values are presented redundantly. This observation is valid for the other figures that show the results. This observation is valid for the other figures that show the results.
Line 386 - Figure 7 and line 522 - Figure 13: It is imperative to improve the presentation of the figure.
Round 2
Reviewer 2 Report
I like the modifications in this version. Minor changes: 1. In table 2, usually use asterisk/asterisks to indicate statistical significance instead of color.
2. In all forest plot, it is better to show sample size for each study.
Reviewer 3 Report
Thank you for taking the time to consider the potential problems I have raised. While I still have concerns about the validity of the results, this has been addressed as a limitation. While there may be a way to do this properly, it could be quite technical and not available in standard software. Given this is noted as a limitation, others may try it in the future, but it is also heartening to note that other reviewers see merit in the findings and given that, future attempts may simply be confirmatory. Additionally, future findings may allow for new methods to be used to account for the dependence and so in that sense this would be an argument for these findings to be presented.
In regards to the small correlations, I didn't mean that they were random noise, rather that if you were to look at any scatterplots with such small correlations, they would simply look like random noise and visually then that there was nothing of interest. That said, a justification for this has been made, and other reviewers don't see this as an issue in this context.
